# AXL-Receptor Targeted 14FN3 Based Single Domain Proteins (Pronectins™) from 3 Synthetic Human Libraries as Components for Exploring Novel Bispecific Constructs against Solid Tumors

**DOI:** 10.3390/biomedicines10123184

**Published:** 2022-12-08

**Authors:** Craig A. Hokanson, Emanuela Zacco, Guido Cappuccilli, Tatjana Odineca, Roberto Crea

**Affiliations:** 1Protelica, Inc., 26225 Eden Landing Road, Suite C, Hayward, CA 94545, USA; 2UCSF, 505 Parnassus Ave., San Francisco, CA 94143, USA; 3Genedata AG, Margarethenstrasse 38, 4053 Basel, Switzerland

**Keywords:** 14 Fibronectin III scaffold, Pronectins, Evolutionary CDR diversity, AXL receptor, bi-specific T cell engager, CD3, solid tumors and metastatic cells

## Abstract

A highly specific AXL-receptor targeted family of non-immunoglobulin, single domain protein binders (Pronectins™) have been isolated from three (3) synthetic libraries that employ the human scaffold of the 14th domain of Fibronectin III (14FN3) and evolutionary CDRs diversity of over 25 billion loop sequences. The three libraries, each containing diversity in two loops, were designed to expand upon a human database of more than 6000 natural scaffold sequences and approximately 3000 human loop sequences. We used a bioinformatic-based approach to maximize “human” amino acid loop diversity and minimize or prevent altogether CDR immunogenicity created by the use of mutagenesis processes to generate diversity. A combination of phage display and yeast display was used to isolate 59 AXL receptor targeted Pronectins with KD ranging between 2 and 100 nM. FACS analysis with tumor cells over-expressing AXL and the use of an AXL knock-out cell line allowed us to identify Pronectin candidates with exquisite specificity for AXL receptor. Based upon several in vitro cell-based tests, we selected the best candidate, AXL54, to further characterize its in vitro cancer cells killing activity. Finally, AXL54 was used to produce the first bi-specific T cell engager protein (AXL54 [Pronectin]-linker-scFV CD3), a “new in class” protein for further testing of its anti-tumor activity in vitro and in vivo.

## 1. Introduction

Protein engineering platform technologies developed in the last few decades are accelerating protein drug discovery and development tremendously [1,2]. Although the initial focus has been the optimization of the IgG structure [3,4,5] due to the medical and commercial importance of monoclonal antibodies (MAbs), new non-immunoglobulin protein libraries, based upon “human scaffolds” and synthetic loop diversity, have been produced by mutagenesis methods and called generally “antibody mimics” as source of potential therapeutic drugs [6,7,8]. Powerful phage [9,10,11] and yeast display [12,13] have facilitated the screening and selection of numerous protein analogs with specificity and selectivity similar to those of Mabs [14]. These Mabs “mimics” have showed great potential in delivering therapeutic benefits [15]. Among the benefits provided by small binding proteins are reducing the manufacturing cost, increasing tissue permeability, stability, safety and efficacy, and the additional benefits of binding to hidden tumor membrane receptor targets that may ultimately affect cancer cells targeting and tumors treatment. Monobodies [16], Adnectins [17,18] and Centyrins [19], all based upon the human scaffold of the 10th domain of Fibronectin III, have provided the ground for developing several clinical candidates for cancer treatment and viral blockade whether used as single protein or in combination with effector moieties and cells of the immune system [20,21,22].

The concept of “the smaller the better” in protein engineering and drug development is still very attractive among biopharma companies. The successful use of single domain proteins as effective therapeutics has been hampered, however, by both pharmacokinetic attributes and potential immunogenic proprieties [17,23]. In the case of FN-III based single domain proteins, their potential loop immunogenicity together with their short residence in the body circulatory system are still areas of great concerns [24]. The potential of using single, hypo-immunogenic protein domain as homo and hetero-multimers or in combination with additional effector proteins still remains unexplored in clinical applications. By the combination of synthetic DNA, intelligent mutagenesis and bioinformatics, human, non-immunoglobulin, synthetic scaffold-based libraries of Pronectins [25] have been produced to optimize the “druggability” of these new classes of synthetic proteins.

Pronectins™, derived from the 14th module of Fibronectin III, were conceived with the precise intent of optimizing CDR (loops) diversity and scaffold stability, based upon a human database created by a bioinformatic analysis of amino acids positional frequency and loop length analysis. A large, “evolutionary” loop diversity of 25 billion analogs has been produced via the synthesis of 3 libraries of 14FN3 variants bearing diversity in two of the three loops [26]. The difference from the more established 10FN3 based libraries is two-fold: we selected the scaffold based upon an extensive bioinformatic analysis of more than 6000 structures based upon structural stability and, equally important, we designed a large human CDRs diversity in the three loops based upon a thorough and detailed analysis of the loop length and amino acid diversity and frequency in a database created from more than 3000 human sequences. This latter approach was meant to eliminated or minimize the possibility of loop immunogenicity created by alternative mutagenesis methods [27]. The detailed synthesis of these libraries has been reported previously [26]. As one of the first applications against solid cancer and metastatic cells, we embarked in the screening and selection of the above libraries for Pronectins highly specific against the AXL membrane receptor.

Axl belongs in the Tyro3, Axl, MerTK (TAM) subfamily of the tyrosine kinases. Axl consists of two immunoglobulin-like (IgG) domains, two fibronectin domains, and a receptor tyrosine kinase [28,29]. Axl receptor is over-expressed in several tumor types and well accepted as a clinical target for metastatic cancer cells [30,31]. The Gas6 (natural ligand) /Axl signaling promotes tumor cell survival, proliferation, migration, invasion, angiogenesis, therapeutic resistance, and immune evasion. Gas 6 and Axl are expressed by host stromal cells, including endothelial cells, fibroblasts, osteoblasts, monocytes, platelets, natural killer (NK) cells, dendritic cells (DCs), and macrophages. As AXL receptor has been widely implicated in metastatic cell cancer, numerous antagonists, whether small molecules and/or Mab and CART cells have been proposed to mainly control the activation of the tyrosine kinase transmembrane signal that leads to the activation of numerous neoplasm mechanisms that leads to cancer [32,33].

Highly specific tumor receptors Pronectins would be instrumental for the design of a novel class of bi-specific T cell engager proteins and other synergistic combinations like in ADC, CAR-T and NK cells for developing new potential therapeutics against both blood cancer and solid tumors [34,35,36].

## 2. Materials and Methods

### 2.1. Pronectin Universal FN3 Libraries

The Pronectin Universal FN3 Library is based on the 14th domain on human fibronectin. Three two-loop libraries designated as BC-DE, BC-FG and DE-FG were assembled by polymerase cycling assembly and cloned into a phagemid vector. The oligonucleotides share fifteen base pairs of overlapping homology with its reverse orientated neighboring oligo. In total the fibronectin gene is assembled with eight oligonucleotides ranging in sizes from 39 to 63 base pairs. Oligonucleotides 1, 2, 4, 6, 8 are the framework of the fibronectin gene where no mutations are designed. The three loops of the fibronectin are encoded in oligos 3 (BC), oligo 5 (DE), and oligo 7 (FG). All eight oligos were mixed to assemble the fibronectin gene by PCA. 

### 2.2. Phage Library Generation

To display the FN3 libraries, the pBluescript-based phagemid vector was built with an N-terminal signal sequence from the *E. coli* DsbA gene for periplasmic export, a c-myc epitope tag and a hexahistidine tag, the C-terminal domain (aa250–406) of the M13 p3 protein, and an amber stop codon. Expression of the fusion protein is under the control of an inducible lac promoter. The three two-loop Pronectin libraries were cloned into the BamH I and Xho I sites on the MCS. The FN3 libraries ligated DNA was electroporated into TG1 *E. coli* cells (Lucigen) and the cloning efficiency was determined by plating serial dilutions on LB plates containing 100 µg/mL carbenicillin. The total number of variants was calculated as BC-DE (9.71 × 10^9^, BC-FG (9.95 × 10^9^) and DE-FG (2.21 × 10^10^). After sequencing, the total number of variants of the combined three libraries was determined to be 25 billion variants. The libraries were selected and expanded on 500 cm^2^ LB-agar-1% glu-carb plates overnight. The cells were scraped into 10–25 mL 2YT + Glu + carb + 15% glycerol and frozen. The three library glycerol stocks were infected with M13K07 helper phage at an MOI = 20 and grown overnight in 2YT-100 µg/mL carb- 25 µg/mL kan- 0.1 mM IPTG. The phage was precipitated with a 1/5 volume 20% PEG/2.5M NaCl. The titers (cfu/mL) of the purified phage library was determined by dilution series on LB agar plates with 100 µg/mL of carbenicillin. The titer of each library stock was BC-DE (1.9 × 10^13^ cfu/mL), BC-FG (1.5 × 10^13^ cfu/mL), and DE-FG (1.5 × 10^13^ cfu/mL).

### 2.3. Pharmacokinetic Study

To facilitate the detection of the Pronectin in serum, the protein was biotinylated via an engineered C-terminal cysteine residue using Maleimide PEG_2_-Biotin (Pierce, Waltham, MA, USA) as per instructions. Female BALB/c mice were randomly divided into 8 groups (n = 4). The Pronectin groups were each dosed intravenously via the tail vein at 5 mg/kg. The control group received PBS at equal volume to the Pronectin groups and blood was drawn at 0 h and 24 h. Blood samples for the Pronectin groups were taken at 8 timepoints (0 h, 0.08 h, 0.25 h, 0.5 h, 1 h, 2 h, 3 h, 4 h). Blood was collected into tubes without anticoagulant and put on ice for 20 min. Tubes were centrifuged and serum was collected. Serum samples were frozen within 15 min at −80 °C. The biotinylated Pronectin was detected by an ELISA assay with a detection limit of 0.24 ng/mL biotinylated Pronectin in mouse serum. First the 6XHis-tagged Pronectin was captures from the serum samples on Ni-NTA-coated microtiter plates (Qiagen). Samples were diluted in 1% BSA/1X PBS, added to the plate and incubated 2 h. at room temperature on a shaker. The plate was washed four times with 0.05% PBST and incubated for 45 min. with SA-HRP at a 1:2000 dilution (Sigma A S5512-1 MG, St. Louis, MO, USA). The plate was washed 3 times with 0.05% PBST and washed an additionally three times with 1X PBS. A total of 100 uL of the QuantaRed™ Chemiflourescent HRP Substrate (Thermo Scientific #15159, Waltham, MA, USA) was added and mixed. The plate was immediately read on a microtiter plate reader for relative fluorescence units (RFU) at excitation 535 nm and emission 595 nm. Purified recombinant biotinylated Pronectin protein at concentration ranging from 2.5 ng/mL to 4 pg/mL were diluted in 1× PBS containing 0.2% mouse serum for the standard curve. From the concentration vs. time plot, the area under curve (AUC) and PK parameters were calculated.

### 2.4. Panning and Screening

Three milligrams of Protein A magnetic beads (Thermo Fisher Scientific, Waltham, MA, USA) were bounded with either recombinant Human Axl Fc Chimera (R&D Systems, Minneapolis, MN, USA) protein or Recombinant Human IgG_1_ Fc (R&D Systems). For Round 1 of phage panning the Axl target protein was 0.5 µM, Round 2: 0.2 µM, and Round 3: 0.1 µM. For all three rounds of panning the Fc target protein for depletion was 0.5 µM. The proteins were bound to the beads in 400 µL of PBS-0.02 Tween-20 and rotated for 15 min, followed by 30 min of blocking with 1× PBS-2.5% milk). Each of the three libraries were panned separately. For round one 1.0 × 10^13^ phage was mixed with the Fc beads in 1× PBS-2.5% milk for 1 h at RT with rotation to depleted out Fc phage binders. The phage supernatant depleted of non-specific binders were transferred to the AXL bound beads and mix for two hours at RT. The beads were washed four times with 1 mL 1× PBS, 0.1% Tween-20, then one wash of 1× PBS-2.5% milk, and a final wash of 1X PBS. The phage was eluted off the beads for ten minutes in 100 µL of 0.1M HCl and neutralized in 900 µL 1M TRIS-HCl, pH 7.4. The eluted phage was amplified in TG1 *E. coli* cells and titered for use in the following round of panning. The phagemid bacterial clones from the tittering agar plates were picked into 96-well block plates aliquoted with 500 µL 2YT-carb-1% glucose per well and grown overnight at 37 °C. An inoculum of 3 µL of overnight culture was added to 200 µL 2YT-carb-glucose and grown until an OD_600_ = 0.4–0.5. The M13K07 helper phage was added at an MOI = 20. The plate was incubated for 30 min at 37 °C with no shaking and then with shaking for an additional 30 min. The well volume was increased to 1 mL with 2YT media-carb-25 µL/mL kanamycin + 0.1 mM IPTG and grown overnight at 30 °C. The phage was harvested by centrifugation. Phage ELISAs were conducted on 96-well flat bottom Immuno plates Maxisorp (Nunc) coated with the AXL-Fc and Fc target proteins in 1.5 µg/mL Bicarbonate coating buffer. The phage was detected using Anti-M13-HRP conjugate (GE Healthcare, Chicago, IL, USA) and TMB substrate (Pierce). The Specific to Non-specific binding ratio was calculated and clones with a ratio 10 and above were submitted for rolling circle amplification and Sanger sequencing with M13 Reverse primer.

### 2.5. Yeast Display Expression

TG1 cells were infected with the Round 2 phage from all the Round 2 phage panning and cultures grown overnight. The phagemid DNA was isolated with a Qiagen Maxi prep column. The Pronectin sequence was excised from the phagemid vector and cloned into pYD1 vector with BamH I and Xho I. The pYD1 vector backbone and Pronectin fragment were gel purified using the stain methylene blue and ligation were set up using 5 µg backbone DNA and 600 ng fragment DNA and incubated for 5 min at 65 °C and 5 min ice. Then, 10× ligase buffer and T4 ligase (NEB) were added and the ligation was incubated overnight at 16 °C. The ligated DNA was transformed by electroporation into DH10B Electro-competent cells (NEB) and grown overnight in culture. Plasmid DNA was purified by Maxi prep column. EBY100 electro-competent yeast cells were prepared by growing 250 mL of yeast at 30 °C to an OD_600_ = 1.3–1.5 and 2.5 mL of TRIS/DTT Buffer (1 M Tris p 8.0, 2.5 M 1,4-dithiothreitol) to the shaking culture for 15 min. The cells were washed three times with decreasing volumes ice cold Buffer E (10 mM Tris pH 7.5, 270 mM sucrose, 1 mM MgCl_2_) and kept on ice. A total of 80 µL of cells were mixed with 6 µg of plasmid DNA in a chilled 2 mm gap electroporation cuvettes. After electroporation of the cuvettes, the cells were recovered in YPD media for 1 h. The cells were then pelleted and resuspended in YNB-CAA glucose media. The transformation was titered on Minimal Dextrose plates. After passaging the cells 2–3 times, expression was induced in YNB + CAA galactose media for 24 h.

### 2.6. Yeast FAC Sorting

10^8^ yeast cells and 10^6^ cells for the controls (unstained and two single color controls) were stained for sorting. The cells were washed with 1 mL of 1× PBS-0.1% BSA and mixed for binding with 2.5 nM AXL-Fc protein at RT. The cells were washed with PBS-1%BSA and 5 µL Anti-V5-FITC and 5 µL Anti-Fc-PE were added and the tubes were incubated in the dark on ice for 20 min. The cells were washed and pelleted. The cell pellets were kept on ice and resuspended in 1 mL PBS-BSA just before FAC Sort #1. A BD FACS Aria Flow Cytometer was used to sort and collect the top 0.5–1% binding cells in 1 mL of YNB + CAA + Pen/Strep. An additional 5 mL of media was added and the plate incubated at 30 °C with shaking for two days. Sort #2 repeated the staining and FAC sorting and collected individual cells in 96-deep well block plates.

The yeast plasmid DNA was purified by the Zymoprep Yeast Plasmid Miniprep II Kit (Zymo Research, Irvine, CA, USA) and was transformed into NEB 5-alpha competent *E. coli* cells. The plasmid DNA was isolated by QIAprep mini-prep Kit (Qiagen, Valencia, CA, USA) and send out for sequencing with T7 promoter primer.

### 2.7. Soluble E. coli Protein Expression

Lead candidates were cloned into the pET-24a vector by PCR amplifying the unique Pronectin sequences with Nde I and Xho I restriction sites. Expression was done in 50 mL Overnight Express LB Media (Novagen, Madison, WI, USA) and 100 µg/mL Kanamycin. The cells were harvested and the cell pellets lysed. The protein was purified by Talon Spin Columns (Clontech, Mountain View, CA, USA) and buffer exchanged into 1× PBS using a Zeba desalting column. Protein concentration was determined in a BCA Assay (Pierce).

### 2.8. Bio-Layer Interferometry (BLI)

The Pronectin lead candidates were analyzed by BLI on the Forte Biosystem Octet QR. The AXL-Fc target proteins were loaded onto Protein A sensors at 25 µg/mL. Pronectins were added at 10 µg/mL. The K_on_, K_off_ and K_D_ for the Pronectins were calculated by the software.

### 2.9. FACS-Based HTS Method

Sixty high affinity AXL binding Pronectins were identified by screening 1 µM Pronectin on MDA-MB-231 (Axl expression) and 293 cells (No Axl expression) using a Novocyte NovoSampler Pro flow cytometer (Acea Biosciences). The Pronectins were ranked by ratio MDA MFI vs. 293 MFI. In a U-Bottom Polystyrene plate, 0.1 × 10^6^ cells/well were seeded in 100 µL of FACS buffer (1× PBS + 5% FBS) for each cell line. Pronectin was added to each well at 1 µM and plates were incubated at 4 °C for 30 min. Plates were spun down at 1500 rpm and washed two times with FACS Buffer. Then, 1 µL of Anti-HIS Alexa Flour 4888 (R&D Systems) and 1 µL of Anti-hAXL APC-conjugated Antibody (R&D Systems) were added for 30 min at 4 °C. The samples were spun down at 1500 rpm for 3 min and washed two times with FACS Buffer. Finally, 100 µL of FACS buffer was added and the cells were analyzed by flow cytometer. 

### 2.10. MDA-MB-231 Knock-Out Cell Line

The AXL gene expression was knocked-out of the MDA-MB-231 cell line using the GeneArt CRISPR Nuclease Vector with OFP Reporter Kit (Thermo Fisher Scientific). The designed single-stranded DNA Oligonucleotides were:

AXL-1F: CACCCCTTATCACATCCGCGGTTT and AXL-1R:

CGCGGATGTGATAAGGGGTGCGGTG;

AXL-2F: ACATTAGTGCTACGCGGAATGTTTT and AXL-2R:

ATTCCGCGTAGCACTAATGTCGGTG;

AXL-3F: TTATCACATCCGCGTGGCATGTTTT and AXL3-R:

ATGCCACGCGGATGTGATAACGGTG.

A 6-well plate with 70% MDA-MB-231 cells confluency was transfected with 3 mg/mL PEI, pH 7.0 and 3 µg of the CRISPR/Casp vector. The population was enriched by FACs for staining with the human CD4 Alexa Flour 488-conjugated Antibody (R&D Systems)

### 2.11. AXL54-CD3 Bi-Specific Pronectin

The FN3 domain AXL54 and the scFV SP34 CD3 were cloned into the vector pmaxCloning (LONZA) through the EcoRV site. The AXL54 and SP34 were synthesized as G-Blocks (IDT) and cloned into the vector using Gibson Assembly Master Mix Kit (NEB).

AXL54 G-Block:

gtttaaacaagcttGAATTCtctagagattgaGCCACCATGGATATGCGAGTGCCAGCGCAACTGCTGGGTCTTCTGCTTCTTTGGTTGTCTGGAGCACGCTGTAACGTCTCACCACCTAGACGGGCGCGAGTTACGGATGCTACCGAAACTACCATAACAATTCGATGGGAGGGCAGAGGTGTGATAGGGTTCCAGGTAGATGCTGTACCGGCAAACGGACAGACCCTATCCAGCGGACAATTAAGCCTGACGTGAGGAGCTATACTATCACCGGATTGCAGCCGGGTACAGATTACAAGATTTATCTCTATACGCTTCCAGACTTCGGGCGAGGACATTTTACAAGCTCTCCAGTAGTAATAGACGCATCAACTGGGGGTGGAGGATCTGAGCTG

SP36His G-Block:

GGGGGTGGAGGATCTGAGCTGGTCGTGACACAAGAACCATCCCTGACCGTGTCTCCGGGAGGAACCGTTACACTGACTTGTCGATCTAGCACAGGCGCTGTTACGACGTCTAACTATGCGAACTGGGTGCAGCAAAAGCCAGGCCAAGCACCTCGCGGCCTCATAGGCGGCACGAACAAGCGGGCACCCGGTACGCCTGCTAGATTCTCCGGGAGCCTGCTTGGAGGGAAGGCTGCACTTACGTTGAGTGGTGTTCAACCCGAGGATGAAGCAGAATACTACTGCGCTCTTTGGTACTCTAACTTGTGGGTTTTCGGAGGGGGCACCAAGCTTACGGTCCTCGGGGGTGGTGGCTCCGGCGGAGGAGGATCTGGAGGTGGCGGTTCCGAAGTTCAACTTCTGGAATCCGGTGGCGGTCTCGTGCAACCGGGAGGTTCACTCAAATTGTCTTGTGCAGCCTCCGGCTTTACATTCAACACCTACGCAATGAACTGGGTGAGACAGGCTCCCGGCAAAGGACTTGAATGGGTCGCGCGGATACGGTCCAAGTATAACAATTATGCTACCTATTACGCAGATAGTGTAAAGGACCGCTTTACGATTTCCAGGGACGACAGTAAGAACACAGCATACCTCCAAATGAACAACCTGAAAACGGAAGACACTGCGGTTTACTACTGTGTCCGACACGGAAATTTCGGGAACTCCTACGTGTCTTGGTTTGCATACTGGGGACAAGGTACACTCGTAACAGTGAGTAGCCACCATCACCATCACCACTGATAAatcctgcagagatctggatccctcgaggctagc

Thirty to forty 15 cm dishes were seeded with 1 × 10^7^ 293-T cells overnight for the transient transfection of the plasmid DNA by PEI. For each plate 40 µg of DNA was mixed in 700 µL of serum free Opti-MEM media (Gibco, Carlsbad, CA, USA). Separately 40 µL of PEI was mixed at 3 mg/mL, pH 7.0 in 700 µL of serum free Opti-MEM media. The two mixes DNA/PEI complex were combined and incubated for 15 min at RT. The mixture was pipetted dropwise directly into the dishes with cells, incubated for 1 day at 37 °C, 5% CO_2,_ and then plates were transferred to 32 °C, 5% CO_2_ for 5 days. Media was collected and and filtered through a 0.45 µm filter.

### 2.12. Protein Purification

The collected media with the AXL54-x-CD3 protein was purified by HIS GraviTrap™ Kit (GE Healthcare). The column was packed with Ni Sepharose Fast Flow resin. The eluted protein was concentrated to a 5 mL volume using an Amicon Ultra-15 Centrifugal Filter Unit 3 KDA MWCO (Millipore, Burlington, MA, USA). The concentrated protein was purified by SEC on an ÄKTAprime FPLC (Amersham Biosciences, Amersham, Buckinghamshire, UK). The column was a HiLoad 16/60 Superdex-200 prep-grade (GE Healthcare). The column was calibrated using the Gel Filtration Calibration Kit LWM (GE Healthcare). The column was equilibrated in 400 mL Gel Filtration Buffer (1× PBS, 10 mM Sodium Phosphate, pH 7.4 and 150 mM NaCl) at 0.5–1.0 mL/min overnight. The sample loop was washed out with 25 mL Gel Filtration buffer and 5 mL sample was loaded into the loop. The column ran at a flow rate of 0.5 mL/min, 200 mL elution volume, 4 mL fractions. A total of 10 µL of protein was run on SDS-PAGE gels and stained with SimplyBlue SafeStain (ThermoFisher Scientific) to identify the fractions with the protein. The peak fractions were pooled and concentrated to 10 mL with Amicon MWCO 3 kDa. The purified protein was dialyzed overnight in 4 L Storage Buffer (50 mM sodium acetate buffer, pH 4.5, 500 mM NaCl) at 4 °C using SnakeSkin Dialysis Tubing (ThermoFisher Scientific). Protein was stored at −80 °C.

### 2.13. T-Cell Dependent Cellular Cytotoxicity (TDCC) Assay

In 90 µL per well, 10,000 MDA-MB-231 target cells and 100,000 PBMC effector cells were seeded into 96-well white, TC-treated plates. The E:T ratio was 10:1. The AXL54-x-CD3 protein was prepared at 10× concentration by adding 10 µL per well in serial dilution (final concentration is 1×). The dilutions started at 100 nM (5–8 fold dilution). The plate(s) were incubated for 48 h at 37 °C. The caspase 3/7 reagent was prepared per manufacturer’s protocol (Promega). A total of 100 µL of reagent was added per well. The plate(s) were covered with foil and the contents of the wells were mixed using a plate shaker at 300–500 rpm for 10 min at room temperature. The plate was incubated for an additional 50 min, under foil at room temperature. Measured luminescence using a luminometer.

### 2.14. T Cell Activation Assay

In a flat bottom TC plate 10,000 target cells (MDA-MB-231) and 100,000 effector cells (PBMC) were seeded in 90 µL. The AXL54CD3 protein was added to the wells in a range from 0.03–500 nM and incubated for 4 h at 37 °C. The cells were harvested and the washed with FACS buffer. The cells were stained with 20 µL CD8-FITC (BD Biosciences) and 20 µL CD69-PE (BD Biosciences) for 30 min and then washed two times with FACS Buffer. The cells were resuspended in 100 µL of FACS Buffer and analyzed by flow cytometry. 

### 2.15. AXL-Fc Competition

The TDCC assay was set-up and spiked with 500 nM of AXL-Fc and human Fc to test that cytotoxicity is decreased by the addition of free AXL-Fc protein. In a flat bottom TC plate, 10,000 target cells (MDA-MB-231) and 100,000 effector cells (PBMC) were seeded in 90 µL media. The AXL54CD3 protein was added to the wells in a range from 0.03–500 nM for sample one. The AXL-Fc and human Fc were added at 500 nM. The cells were incubated for 4 h at 37 °C. The cells were treated with 100 µL of the Caspase-Glo 3/7 Reagent (Promega) by covering plate with foil and mixing on a plate shaker 300 rpm for 10 min. After an additional 50 min of incubation on lab bench, the plate was measure for luminescence with a luminometer.

## 3. Results:

### 3.1. Bioinformatic Approach for 14FN3 Selection as a Scaffold

In choosing a fibronectin III scaffold, a bioinformatic consensus approach was implemented. Fibronectin Type III (FN3) sequences were downloaded from the PFAM and PROSITE protein database. The initial dataset of 9321 protein sequences contained FN3 modules 1–16, tenascin, cytokine receptors, chaperonins, carbohydrate binding domains, tyrosine kinases and phosphatases. Repeat sequences were removed and the resulting data set contained nearly 6000 non-redundant aligned FN3 polypeptide sequences. The data set was further reduced nearly 3000 sequences by using only human diversity with a BC loop of 11 amino acids (626 sequences), human DE loop of 6 amino acids (1210 sequences), and human FG loop of 9 amino acids (1151 sequences). Using this alignment and 10FN3 (94 amino acids) as the reference, the three loop amino acid sequences were identified. The amino acid designation for the loops are BC (21–31 a.a.), DE (51–56 a.a.), and FG (76–86 a.a). 

In choosing a FN3 domain to build libraries, we decided on the 14th domain of FN3 Type III. In our analysis of all the FN3 domains, the 14FN3 most closely resembles the three-dimensional characteristics and retains key residues in the framework of the proven 10FN3 scaffold. The first step was to align the three-dimensional structures from the PDB database of the available human FN1–FN16 modules with the 10FN3 structure and look for the sequences showing minimum spatial divergences from 10FN3. Then, the residue identity and conservation at key positions, such as the boundary between the beta strands and the loop regions, were compared to 10FN3. In the end, module 14FN3 had the greatest degree of three-dimensional and sequence resemblance to 10FN3 (Figure 1A).

The loops length distribution were analyzed to discover any loop length preferences. The BC and FG loops have a bias towards loop length 11, 14, 15 a.a. and 8 and 11 a.a., while the DE loop strongly favors only a length of 6 a.a. (Figure 1B). Next a positional amino acid frequency analysis was performed at each loop position using a set of human FN3 modules to determine the conserved, semi-conserved, and variable positions. A conserved (Fixed) or semi-conserved position are, respectively, defined as 50% one amino acid at a position and 40% for 2–3 amino acid at a position. A variable position has no amino acid representing more than 20% of the represented set (Figure 1C). The resulting sequence profile establishes the frequency of occurrence of the amino acids at each position for each loop length. For the libraries we chose a cut-off of greater than 5% to incorporate in the library diversity design (Figure 1D). To reduce the eventual library size each amino acid is carefully considered for its physical properties and redundant characteristics. Thus, not all the identified amino acids are in the library design. Because the library was created using degenerate oligos with mixed bases, they were designed to limit the amount of included diversity that was not in the identified human FN3 amino acid analysis (Figure 1E).

### 3.2. Early Pronectin Universal Fibronectin Libraries Summary

The fibronectin phagemid libraries were assembled by polymerase cycling assembly (PCA) using eight oligonucleotides ranging in sizes from 39 to 63 base pairs and cloned into a modified pBS KS (-) phagemid vector. The vector was designed for monovalent display of the protein through the M13 attachment protein pIII fused to the C-terminus of the Pronectin (Figure 2A,B). The Pronectin library was transformed in TG1 cells and stored as glycerol stocks (Figure 2C,D).

The two loop 14FN3 libraries were successfully tested against several protein targets like VEGFR2, FZD2 and FZD4 (unpublished results). Each of the lead Pronectins discovered against these targets showed specific binding with no cross reactivity in in vitro ELISA assays and FACs assays and several of the lead Pronectins blocked the natural ligands from binding to their receptors. The thermo-stability of the Pronectins were analyzed in a Thermoflour assay using ABI Prism 7700 Sequence Detector for RT-PCR. Most Pronectins with low nanomolar binding fell into the melting point temperature range of 65–75 °C. Extensive loop modeling of the 14FN3 Pronectins was done using MODELLER [37,38] and binary docking was predicted using PatchDock and FireDock [39]. The in silico epitope mapping computational analysis of Pronectin binders to target proteins confirm that the Pronectin library is capable of producing binding molecule to multiple epitopes on a given protein. To characterize the in vivo half-life of our Pronectins, we performed a mouse pharmacokinetic study using a biotintylated Pronectin injected i.v. into male BALB/c mice at 5 mg/kg and serum samples were collected at time points from 5 min to 4 hrs post-injection. Serum levels of the Pronectin were determined by ELISA. As expected, Pronectin R2F8, an 11.6 kDa protein, was rapidly cleared from the blood. The area-under-curve (AUC) value was 2316 ng/mL·h, with a clearance of 129.5 L/min/kg and a half-life of 1.5 h (Figure 3). These values are consistent with other small proteins such as single domain antibodies and 10FN3 proteins. In addition, since unmodified 14FN3 proteins are well below the renal exclusion limit of 60 kDa, a rapid clearance rate was expected. To reduce this rapid clearance, one can use a number of post-translation modifications, such as PEGylation which has successfully reduce the clearance rate of 10FN3 proteins [22]. Alternatively, the rec. DNA fusion of Pronectins to other functional domains may provide longer half-life in blood.

From the above studies, Pronectins were determined to be a viable non-immunoglobulin single domain best suited for applications as a bispecific protein. In the following described work, the best AXL receptor ligand (AXL-54) was therefore chosen as the target for a bispecific protein T cell engager for cells of solid tumors.

### 3.3. Discovery and Characterization of AXL Pronectins

#### 3.3.1. Phage Display Panning

The three Pronectin libraries with two-loop diversity were cloned as phage libraries and panned for binding to the extracellular domain (ECD) of the AXL receptor (AXL Fc Chimera, R&D Systems #154-AL) in a bead-based strategy. Three rounds of panning were completed (Figure 4A) and over 800 recovered clones from the round 2 and round 3 phage panning were analyzed for specific binding to the AXL ECD by phage ELISA. From the Round 2 clones 16% of the phage clones had a specific binding ratio ≥10, while the round three clones had 66% of the clones with a binding ratio ≥10. From the ELISA data 576 clones were sent for rolling circle amplification sequence (Elim Biopharmaceuticals) and 157 unique sequences were discovered. Using Jalview 2.10.1 a phylogenetic tree showing the sequence similarity of the clones was created from the average distance using percent identity on region sequence comparison by ClustalWS alignment (Figure 4B). From the ELISA and sequencing results, 40 Pronectins were moved forward to soluble *E. coli* expression.

#### 3.3.2. Yeast Display Selections

The Pronectins were cloned from the AXL round 2 phagemid DNA into a modified pYD1 vector with the AGA2 mature peptide designed as a C-terminal fusion to the Pronectin. The Pronectins loops are near the N-terminus of the Pronectin and we did not want any interference from the AGA2 mature peptide. The yeast libray was transformed into EBY100 cells by electroporation and selected in SDCAA selection media. Two rounds of yeast display was performed at 25 nM and 2.5 nM by flow cytometry. From round one we sorted out the top 0.5–1% binders into a pool. For round 2 we individual sorted out the top 0.1–0.5% binders to two 96-well blocks. The yeast DNA was mini-prep for 192 clones and sequenced. Sixty-seven unique clones were discovered that were not found in the phage display screening. Twenty of these Pronectin sequences were cloned into the Pet-24a vector for soluble *E. coli* expression.

#### 3.3.3. Binding Affinities

In total 59 Pronectins were expressed in *E.coli* and purified by HIS-tagged purification. A ForteBio Octet QK machine was used to measure the binding affinity of AXL to the recombinant AXL-Fc Chimera recombinant protein by Bio-Layer Interferometry technology (Figure 5A). From the phage discovered clones, we had six Pronectins with a KD below 10 nM. This was 15% of the phage display Pronectins. From yeast display, 7 Pronectins had KD below 10 nM representing 35% of the yeast display Pronectins. The five lowest KDs were for Pronectins discovered by yeast display (Figure 5B).

#### 3.3.4. Screening AXL Pronectins on Cell Lines

AXL expression was tested on 8 cells line (MDA-MB-231, 786-0, SKOV-3, PC-9, A549, PC-3, HepG2 and HEK-293). The top expressing AXL cell line was MDA-MB-231 (abbreviated as MDA-231 on some of the figures) and was chosen as the cell line for further analysis of clones (Figure 6A). HEK-293 was our AXL negative cell line for cell expressing comparison experiments (Figure 6B). The specific binding of each of the 59 AXL Pronectins were tested at 1 uM on MDA-MB-231 and HEK-293 cells in a FACS assay. The Pronectin binding was detected by the HIS tag with an AntiI-HIS-APC antibody. The fifteen lead Pronectins had a ratio greater than 5 for AXL binding to MDA-MB-231 cells vs. HEK-293 cells (Figure 6C). 

Because the specificity of the Pronectin to AXL is of utmost importance we constructed an MDA-MB-231 AXL Knock-out cell line. The knock-out cell line was engineered using the GeneArt CRISPR Nuclease vector with OFP Reporter Kit (ThermoFisher). The top Pronectins were screened for binding to the MDA-MB-231 cells and the AXL Knock-out cell line (Figure 7A). Three AXL Pronectins (AXL44, AXL54, and AXL56) showed none to little non-specific binding to the knock-out cells and were chosen as the lead Pronectins to move forward (Figure 7B–D).

### 3.4. AXL54-x-CD3 Bi-Specific T Cell Engager Pronectin

To test the Pronectin efficacy in preclinical studies, we build an AXL-x-CD3 bispecific proteins to mimic BiTE fusion molecules being used to target T-cells to cancer cells. This represents the first bispecific molecule targeting the AXL receptor with a Pronectin (AXL54) linked to CD3 for engagement of T cells (Figure 8A). The linker (x) is a single unit of Gly4-Ser. We cloned the AXL44, AXL54 and AXL56 as Pronectin fused to the scFV anti-CD3 (SP34) into the pmaxCloning (Lonza). To prevent possible interference with the three binding loops, the Pronectin was cloned 5′ of the CD3 module because the Pronecin’s loops are orientated near the Pronectin’s N-terminus. The pmaxCloning vector allows for constitutive expression of protein in mammalian cells under the immediate early promoter of cytomegalovirus. We engineered in the kappa light chain signal sequence to allow the protein to be secreted out of the cells. The CD3 arm of the AXL bispecific is a humanized version of the muSP34 antibody. The scFV is the light chain, lambda, and heavy chain of SP34 [40]. The scFv SP34 is oriented N-V_L_-(G_4_S)_3_-V_H_-C (Figure 8B). The proteins were transiently expressed in 293-T cells and purified by the HIS-tag. The purified protein yields were 2 mg per liter of supernatant and has the predicted molecular weight of 37.3 kDa. A second round of gel filtration purification was needed to achieve a pure protein. The protein was purified on a size exclusion column using an Äkta FPLC (Figure 8C).

The AXL54CD3 protein oligomeric state was characterized by SEC and the purified protein was retested on MDA-MB-231 cells by FACS for binding to the AXL receptor. Figure 8D shows the FPLC trace of the AXL54CD3 protein run over a HiLoad 16/60 Superdex-200 prep-grade column. The main peak corresponds to the collected 37.3 kDA AXL54CD3 protein (Figure 8D). The protein was tested in three stages (1) after SEC, (2) after dialysis into storage buffer, and (3) after a freeze thaw from −80 °C. The AXL54CD3 protein showed no evidence of aggregation or degradation with no loss of signal in the assay (Figure 8E).

In summary, the binding affinity of the AXL54 Pronectin expressed in *E. coli* was 8 nM, while in a dose–response assay on MDA-MB-231 cells the EC_50_ was 42.6 nM. The EC50 of the bispecific AXL54CD3 was 16.7 nM in the cytotoxicity study, The data is summarized in Figure 9.

### 3.5. AXL-Fc Cytotoxicity Study

To validate the bispecific constructs, we used an in vitro TDCC luminescent assay with great sensitivity for caspase released from dead or dying cells (Caspase-Glow 3/7 Cytotoxicity Assay, Promega). The positive control was an scFvHer2-CD3 molecule [41]. The scFvHer2 was designed from the humanized monoclonal antibody Herceptin (Drug Bank Accession DB00072). The Her2-CD3 control is a bispecific protein containing two scFv. The orientation and design of the protein is scFvHer2 [V_L_-(G_4_S)_3_-V_H_]-G_4_S-scFvCD3 [V_L_-(G_4_S)_3_-V_H_]-6X HIS-tag. The assay was performed with the cell line MDA-MB-231 and PBMC cells at a Target: Effector ratio of 1:10. CHO cells were chosen as the negative control cell line since they do not express the AXL Receptor. In this assay the AXL Knock-cell line we built in house showed high levels of background. We believe the Axl-knockout cell was sensitive to protein aggregates in the AXL-x-CD3 preps. The AXL-x-CD3 Pronectins exhibited cytotoxicity on MDA-MB-231 cells with EC50 ranging from 8 to 17 nM (Figure 10A). The Her2-CD3 positive control had an EC50 of 15 pM. On the CHO cells the AXL54 clone showed minimal cytotoxicity (Figure 10B). Next, we tested the functionality of each arm of the bispecific AXL54-x-CD3 molecule to prove that the results of the cytotoxicity assay are due to an effective molecule.

### 3.6. CD69 Activation Assay

The functionality of the CD3 arm of the bi-specific Pronectin was confirmed by a CD69 Activation Assay with the AXL54-x-CD3 Pronectin titrated from 500 nm to 4 nM on PBMC and MDA-MB-231 cells. T-cell activation was detected by staining the cells for CD8 and CD69. The CD8+ cells showed an increase in CD69 expression (Figure 11A). CD69 is an early activation antigen on T-cells that has been developed as a marker for FACS assays to detect T-cell activation. With the CD3 arm determined to be functional, we tested the AXL arm next. (The AXL54CD3 is the subject of current investigation which will included further testing that the bispecific Pronectins do not activate T-cells in the absence of target cells).

### 3.7. AXL-Fc Competition Assay

The AXL Pronectin arm specificity was confirmed by an AXL-Fc competition assay with MDA-MB-231 cells and PBMC cells. In this TDCC assay, the addition of the AXL-Fc protein to the AXL54-x-CD3 protein decreased cytotoxicity due to the neutralization of the AXL-x-CD3 Pronectin and prevention of the Pronectin binding to MDA-MB-231 cells. The addition on recombinant human IgG1 Fc protein with the AXL54-x-CD3 did not prevent cytotoxicity in the assay (Figure 11B). These assays confirm that the AXL5-x-4CD3 is a functional lead candidate that specifically binds to the AXL receptor on tumor cells and can activate T-cells.

## 4. Discussion

We have shown here that a 14FN3 based library of 25 billion Pronectin variants is a viable source of non-antibody protein binders that can recognize tumor cells receptors, like AXL tumor receptor, in a very specific and selective way. Pronectins are similar in structure to other Fibronectin Type III 10FN3 domain (Monobodies, Centyrins, etc.). These scaffolds are cysteine-free Ig-like ß sandwich fold proteins with three exposed binding loops. Pronectins also exhibit similar binding properties, such as having (1) multiple binding modes to interact with targets, (2) residues outside the diversified loops that can interact with ligands, and most importantly, (3) interactions with epitopes inaccessible to antibodies.

The “human only“ amino acid composition of the Pronectin libraries was designed to minimize immunogenicity. The Pronectin platform combines the fully human 14FN3 scaffold with loop diversity derived from residues found naturally in human FN3 domains. This “evolutionary” approach maximizes the use of amino acids selected by nature over the course of billions of years. In addition, the human 14FN3 fibronectin derived scaffold is based on the abundant extracellular fibronectin protein that is exposed to and tolerated by the immune system. Thus, the Pronectins platform should minimize immunogenic sequence variants. Because the loop diversity is solely generated by a set of naturally occurring amino acids and only preferred loop lengths used, the Pronectin libraries has a higher proportion of stable, functional variants. The incorporation of multiple loop combinations and lengths in the Pronectin libraries allows for optimal recognition of a wide range of targets.

Pronectin library screening uses a two-step selection method that combines the panning of the libraries by phage display and then the subcloning of an enriched library of binders for yeast display. We have shown that only 2 or 3 rounds of selections are sufficient to enrich for a population of highly convergent Pronectin antagonists with low nanomolar binding affinities. Yeast display allows us to quantitatively screen the library using flow cytometry. Since the binding signal can be normalized for variant expression in yeast display, eliminating artifacts due to host expression bias, fine discriminations between binders and different affinities is possible. In addition, the ability to perform initial characterizations in a yeast format allows for the identification of the best clones in terms of functionality. By combining the larger library size and rapid screening of phage display with the quantitative screening of yeast display, we take advantage of the strengths of each system to quickly identify the best Pronectin binders. The use of the MDA-MB-231 AXL Knock-out cell line to screen the AXL Pronectins for non-specific binding was the last and probably most important step for the selection of the lead AXL candidate. The Pronectins from yeast display were screened on the AXL positive MDA-MB-231 cells and AXL negative 293 cells, and 15 out of 59 Pronectins selected as specific binders. However, it was the comparison of the Pronectin binding to the MDA-MB-231 cells to the Knock-out cells that allowed us to distinguish the top three lead Pronectins that showed minimal non-specific binding. From here, AXL54 was chosen as the lead candidate to develop into the bispecific T cell engager protein.

The naked, 10 kDa monomer Pronectin is rapidly cleared from the blood in a mouse. The half-life of a Pronectin is approximately 1.5 h. There are many ways to increase the half-life of a Pronectin. PEGylation has been used successfully to increase the half-life of Monobodies up to 50 h [22]. We have designed the Pronectins to be built as multimer, as hetero and homodimers and trimers. Increasing the size of the protein will slow clearance of the protein. In this study we chose to increase the AXL Pronectin half-life by building a bispecific protein in the manner of a BiTE molecule [42]. The AXL54-x-CD3 molecule has a predicted molecular weight of 37.3 kDa with a substantial longer serum half-life than the monomer AXL54 Pronectin. 

AXL is a promising cancer target because it is expressed in many solid tumor types and is weakly expressed in normal tissues. The over-expression and signaling of the receptor tyrosine kinase AXL by its ligand Gas6 on the surface of several cancer cell lines has been shown to promote their tumor progression, to correlate with their metastatic potential, and can lead to resistance to current cancer therapy. The activation of AXL is over-expression-dependent and regulates proliferation, survival, and migration of cells. AXL has been noted to influence clinically meaningful end points including metastatic recurrence and survival in the vast majority of tumor types [43]. AXL has been implicated in metastasis of numerous cancers, including breast, ovarian, colon, thyroid, lung, liver and others [29]. Several AXL small molecule inhibitors, AXL antibodies for cancer therapy, and soluble decoy receptor molecules have been tested in clinical trials [43]. Additionally, an AXL CAR-T molecule has shown in vivo anti-tumor effects against triple negative breast cancer (TNBC) in an MDA-MB-231 xenograph model [44]. The AXL receptor is a promising target for AXL-targeted cancer drugs. AXL54-x-CD3 is designed as a bi-specific protein that will link AXL-expressing tumor cells and T-cells via the CD3 receptor. 

The AXL54-x-CD3 is highly specific to AXL expressing tumor cells. It showed minimal binding to the MDA-MB-231 AXL knockout cell line. Thus, AXL54-x-CD3 is expected to show low background in in vitro assays. In the T-cell dependent cellular cytotoxicity (TDCC) assay, we saw minimal cytotoxicity against the AXL negative CHO cells. Additionally, the AXL-Fc competition assay confirmed that the AXL54-x-CD3 Pronectin is specifically targeting the AXL receptors on the tumor cells because the AXL-Fc protein was able to bind to AXL54-x-CD3 and neutralize the bispecific molecule, prevent its binding to the tumor cells. The scFv SP34 (CD3) is the humanized version of the primary mouse derived anti-CD3 antibody. The original murine scFv SP34 binding affinity in the low two-digit nanomolar resulted in a suboptimal T-cell immune response [45]. The CD3 arm of the AXL54-x-CD3 bispecific protein was tested in a T-cell activation assay for CD69 induction. CD69 is an early activation marker that the T-cell receptor has been stimulated. At the 4 h time-point, AXL54-x-CD3 molecules was able to stimulate the T-cells in PBMC to express CD69. With both arms of the bispecific protein act XL54-x-CD3 was tested in the TDCC Assay and showed an EC50 of 16.7 nM. In the assay, as the concentration of the AXL54-x-CD3 bispecific protein was increased, more cytotoxicity was measured. The AXL54-x-CD3 bispecific protein is a promising molecule designed to direct the ive, A immune system against tumor cells. The described in vitro assays shows that this protein can be tested for efficacy in solid tumor animal models as “first in class” Pronectin based bispecific T-cell engagers.

## Figures and Tables

**Figure 1 biomedicines-10-03184-f001:**
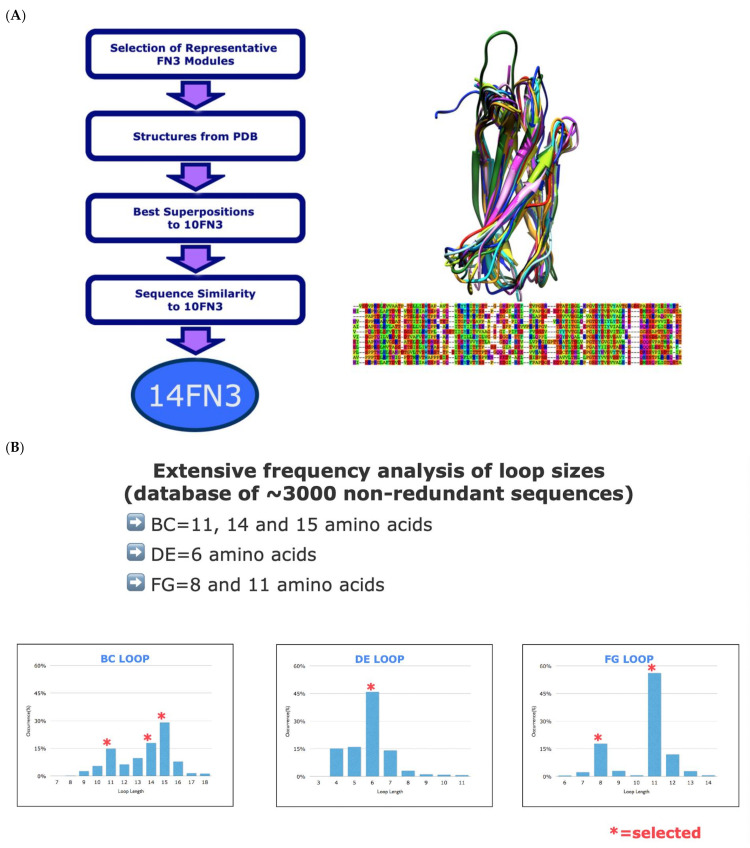
(**A**) Bioinformatic approach for 14FN3 selection as novel scaffold. (**B**) Identification of the most common loop sizes for FN3 domains. (**C**) Alignment of 10FN3 and 14FN3 sequences with * is conserved sequence, : is conservative mutation, and . is a semi-conservative mutation. (**D**) the amino acid distribution for BC loop (11) from human sequences. (**E**) The frequency of each amino acid for the BC (11) loop incorporated into the Pronectin Universal FN3 Library.

**Figure 2 biomedicines-10-03184-f002:**
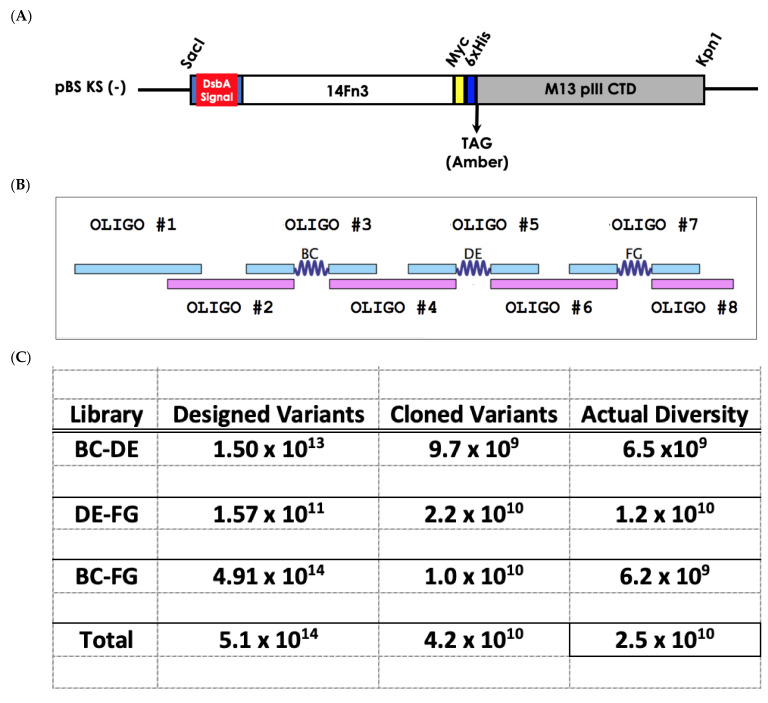
(**A**) Phagemid vector used for the Pronectin Fibronectin Universal Libraries. (**B**) The PCA assembly of the FN3 scaffold using oligonucleotides with 15 base pairs of overlapping homology. (**C**) Summary of the diversity of the three two-loop Pronectin Universal Fibronectin Libraries. The Pronectin Universal Fibronectin Libraries contains 25 billion variants. (**D**) The percentage of each sub-library for the three 2-loop libraries with their varying loop lengths.

**Figure 3 biomedicines-10-03184-f003:**
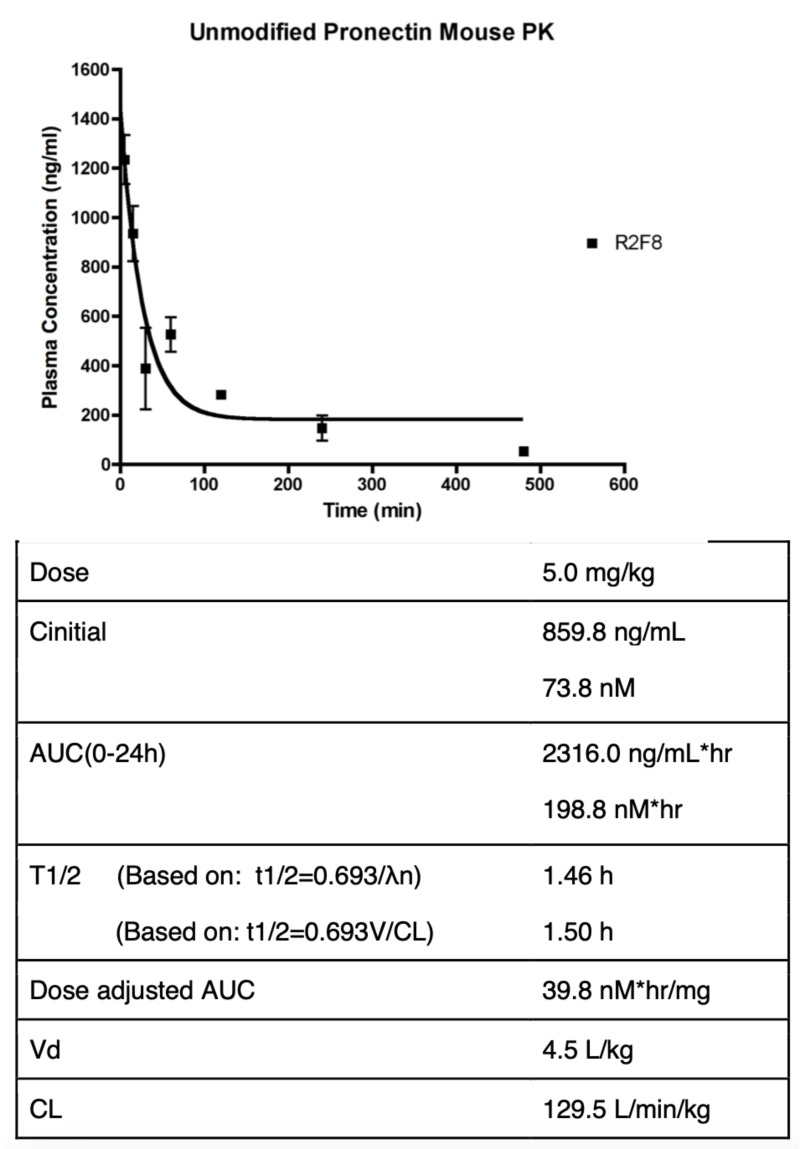
The pharmacokinetic profile and calculated values after administration of 5 mg/kg to BALB/C mice. The half-life of the Pronectin (R2F8) is 1.5 h.

**Figure 4 biomedicines-10-03184-f004:**
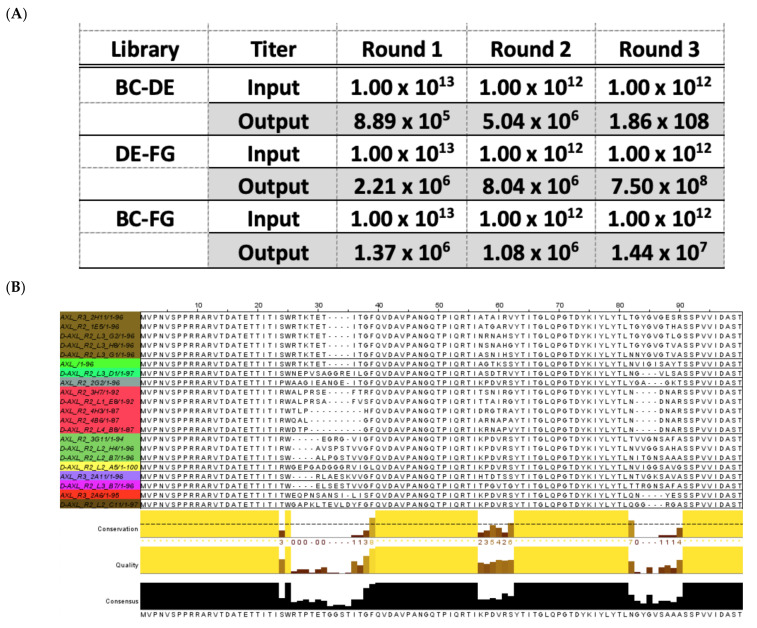
(**A**) Summary of phage display Pronectin Universal Fibronectin Libraries used for panning AXL-Fc Chimera recombinant protein. The four libraries underwent three rounds of panning and enrichment. Input titers are listed as cfu/mL, while output titers are cfu/mL from 0.1 M HCL elution (1 mL). (**B**) A sampling of the sequence space and an example phylogenetic tree used to narrow down the 157 unique AXL Pronectin sequences.

**Figure 5 biomedicines-10-03184-f005:**
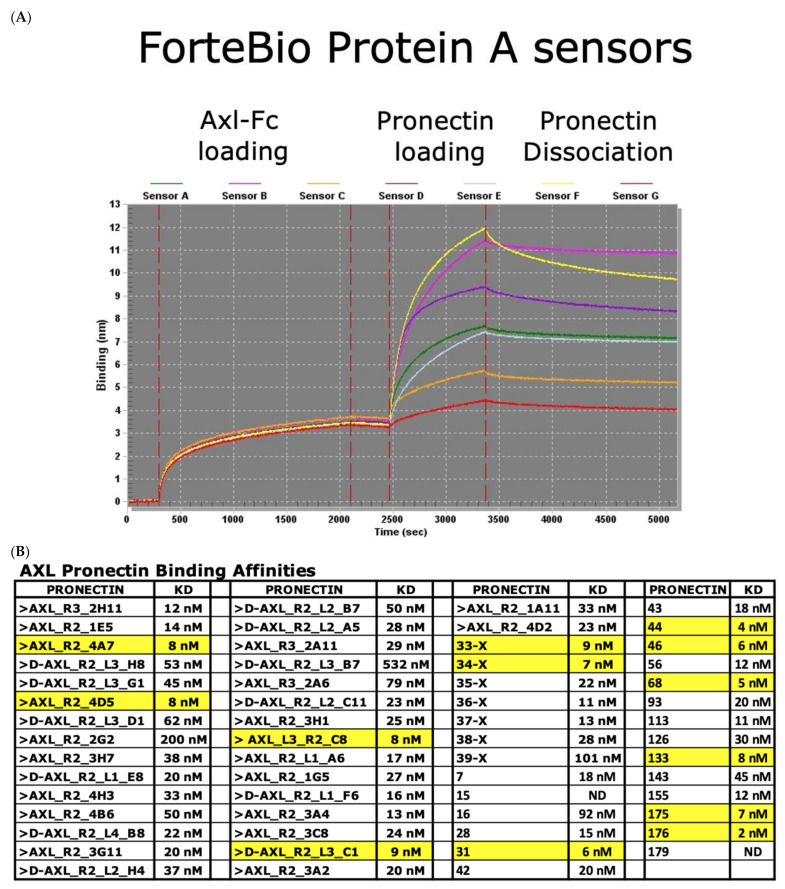
(**A**) Binding affinity curves for lead Axl binder Pronectin. (**B**) The binding affinities KD values for all 59 Axl Pronectins. The KDs < 10 nM are highlighted in yellow. The first 39 Pronectins were discovered by phage display and the additional 20 were new Pronectins discovered by yeast display.

**Figure 6 biomedicines-10-03184-f006:**
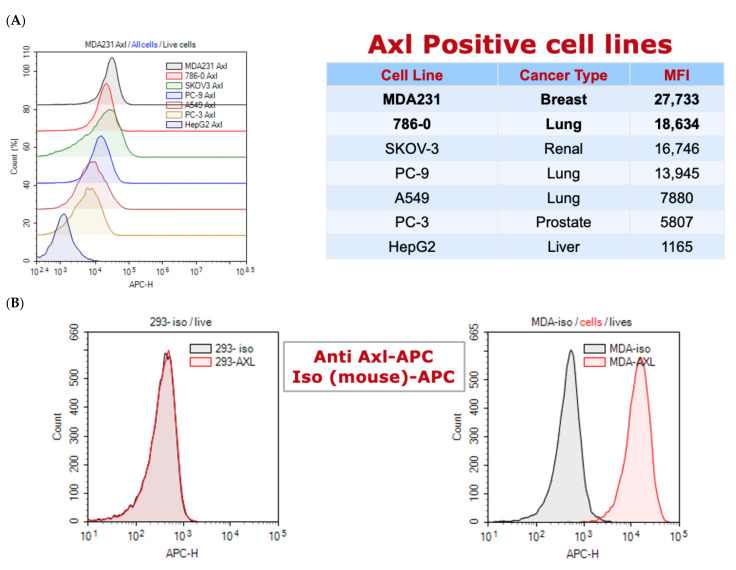
(**A**) Shows the shift in the fluorescence intensity on the population of seven AXL positive cell-lines using an anti-AXL-APC conjugated antibody. MDA-MB-231 has the most AXL expression. (**B**) Show that HEK-293 cells have no AXL expression and that the AXL antibody was specific to the AXL receptor on MDA-MB-231 cells. (**C**) All 59 AXL Pronectins were analyzed by flow cytometry on 293 cells and MDA-MB-231 cells. The ratio of the MFI on MDA-MB-231 cells vs. 293 cells is graphed. The last bar is the binding of soluble wild-type 14FN3 protein and show the non-specific binding levels of the assay. The Pronectins above the 5.0 ratio for AXL binding are the lead candidates.

**Figure 7 biomedicines-10-03184-f007:**
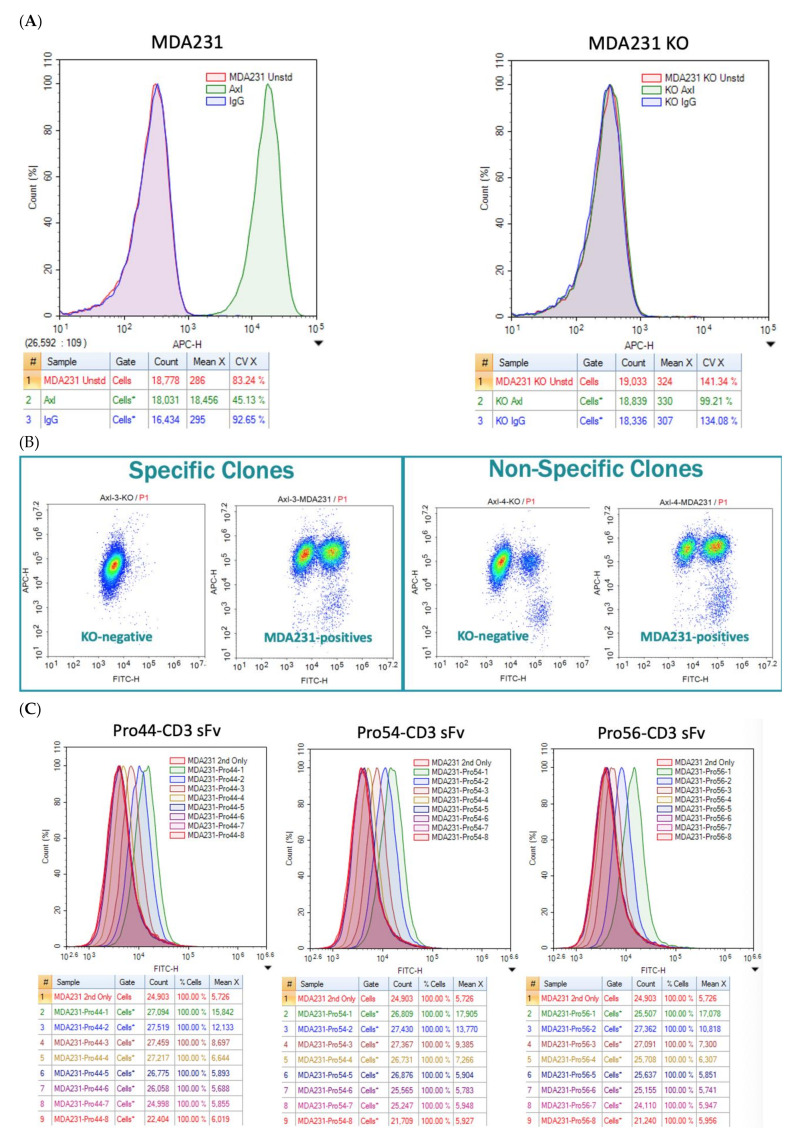
(**A**) Shows specific binding of anti-AXL-APC conjugated antibody to MDA-MB-231 cells and no binding to the AXL Knockout cells. (**B**) Distinguishing specific binders from non-specific binders by FACS. (**C**) Axl Pronectins (44, 54, and 56 starting at 500 nM, 1:4 dilution for 8 points) showing AXL binding titration on MDA-MB-231 cells. (**D**) The three AXL Pronectins show no to little binding on the AXL Knockout cells.

**Figure 8 biomedicines-10-03184-f008:**
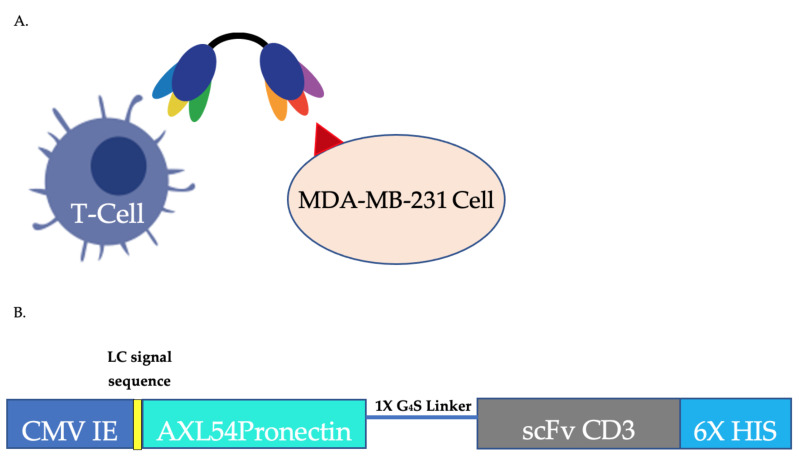
(**A**) Design of the Bispecific AXL54CD3 protein (**B**) AXL54CD3 mode of action (**C**) The reduced SDS-PAGE gel of the purified 37.3 kDa AXL54CD3 protein. Lane 2 is the protein after HIS-tag purification and Lane 3 is after a second size exclusion chromatography step. (**D**) The SEC trace (blue) of AXL54CD3 purified over a HiLoad 16/60 Superdex-200 prep-grade column. (**E**) The functional activity of the protein is tested by FACS on MDA-MB-231 cells after the SEC (green), after dialysis into storage buffer (blue), and after a freeze thaw (brown). Activity remains the same.

**Figure 9 biomedicines-10-03184-f009:**
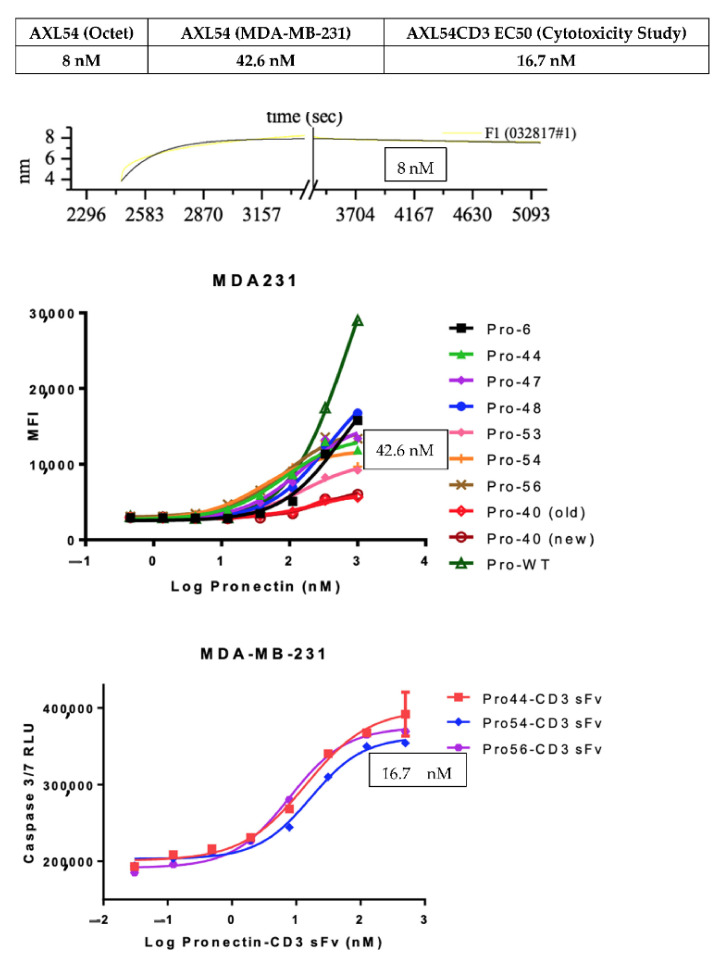
Summary of the Binding affinity of AXL54 Pronectin and the dose-dependent EC50 of the AXL54 on MDA-MB-231 cells and the EC50 of AXL54CD3 in the Cytotoxcity Assay.

**Figure 10 biomedicines-10-03184-f010:**
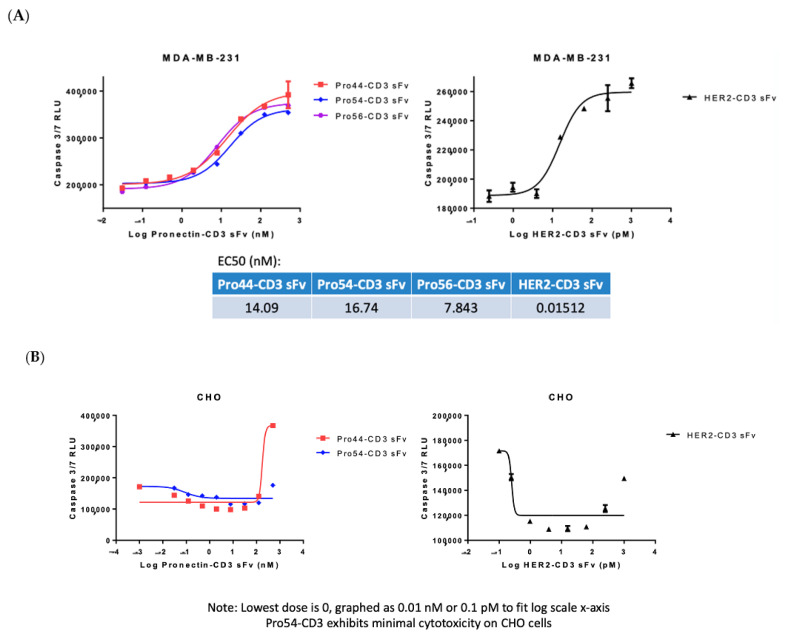
(**A**) Results from the Cytotoxicity Assay of three AXL-CD3 Pronectins on MDA-MB-231 cells. (**B**) Two AXL-CD3 Pronectins and the HERs-CD3 sFv control tested on the negative AXL expressing cell line CHO.

**Figure 11 biomedicines-10-03184-f011:**
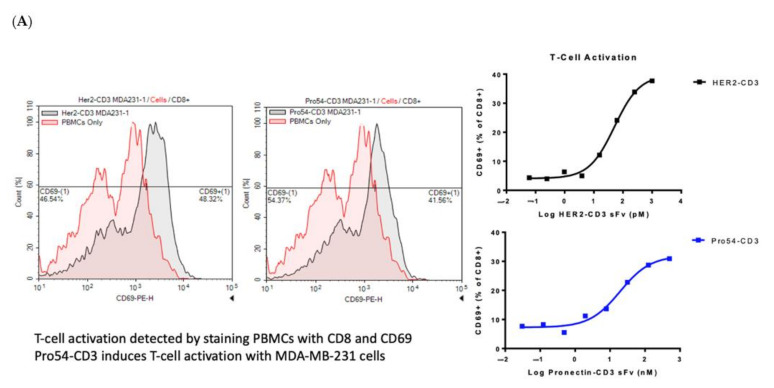
(**A**) Shows the increase in CD69 levels on the CD8+ PBMC cells with the addition of the AXL54-CD3 and the Her2-CD3 bispecific proteins in a CD69 activation assay. (**B**) The AXL-Fc competition assay results showing that the addition of recombinant AXL-Fc protein neutralizes the AXL54-CD3 protein.

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
