# Peer review of "AXL-Receptor Targeted 14FN3 Based Single Domain Proteins (Pronectins™) from 3 Synthetic Human Libraries as Components for Exploring Novel Bispecific Constructs against Solid Tumors"

_biomedicines, 2022, doi:10.3390/biomedicines10123184_

Round 1
Reviewer 1 Report
The authors of the manuscript " AXL-receptor targeted 14FN3 based single domain proteins (Pronectins™) from 3 synthetic human libraries as components for exploring novel bispecific constructs against solid tumors", through screening of phage and yeast libraries supported by bioinformatic analyses, have selected a small domain protein (Pronectins) specifically against the receptor AXL54 using as scaffold the 14th domain of Fibronectin III (14Fn3). they generated a first in class of bispecific T-molecules (scFv CD3-AXL54) that works in vitro.
Although the work is interesting, it appears always inaccurately described.
The figures are all of poor quality and in some cases even incorrectly marked in the text. I suggest to carefully check and replace them.
In the paragraph 3.2 are reported pharmacokinetic data that are not described in the methods. Given the importance of these data, I suggest to conform this part in the experimental procedure.
In addition, I suggest to introduce some details about the design of bispecific gene (i.e. linker, the choose of orientation) and mostly about biochemical characterization regarding both the scFv molecules produced in E coli and bispecific in 293-T.
I suggest to check the oligomeric state after purification to support the functional activity of these proteins in the cells. Is not clear if the reported sds- page, are in reducing conditions or native.
I also suggest to improve the binding data in term of assay description (i.e concentrations), table and figure, at least. Is not clear if the authors report an affinity constant based on the screening or derived from a more detailed dose-response assay. Dose-response assay should be provided to define the affinity constant.
Reviewer 2 Report
This paper by Hokanson, et al. describes the design and construction of a new human like antibody-mimics library based on the 14th domain of human fibronectin III protein (14FN3), as well as the engineering of AXL-receptor binding 14FN3 via phage and yeast display. To minimize immunogenicity from the loop sequences, extensive bioinformatics was employed to design the 14FN3 library. This method of library design and construction is quite novel, and this library has a potential to be used for generating other protein therapeutics. The identified AXL-receptor binder showed low nanomolar target binding ability and a strong potential to function in a Bite format to in tumor therapy.
However, the paper in the current format reads like an unfinished draft and significant editing is necessary. In addition, it is impossible to evaluate the claims of the paper due to the very poor resolution for all the figures. Below are a few specific comments:
1. An important requirement for Bite molecules is their lack of self-aggregation which can trigger pre-mature T cell activation. Thus, size-exclusion chromatography data for AXL54-x-CD3 or AXL54 should be provided in order to support the claim that ‘Pronectins were determined to be a viable non-immunoglobulin single domain best suited for applications as a bispecific protein’ (line 375).
2. The in vivo half-life of Protectin in mice was determined to be 1.5 hours (line 368). However, neither method nor data/figure was presented in the manuscript. These information need to be added to the manuscript.
3. In multiple places, ‘Figure X’ was mentioned (e.g. line 424, 459). Its unclear what is Figure X.
4. Line 457: “The Her2-CD3 positive control had an EC50 of 15 pM”. Is this a HER2 ectodomain fused to CD3 ectodomain? More specifics of the positive control constructs are needed.
Round 2
Reviewer 2 Report
The revised manuscript is greatly improved. I have a few minor comments.
1. Figure 5B is missing.
2. The labeling in Figure 7C, D is confusing. Under each panel there are 8 different samples (-1 through -8) in addition to the control (2nd only). I presume these are the same protein incubated at different concentrations. It would greatly improve the clarity if these concentrations can be included in the figure legend.
3. Figure 9: What’s the difference between MDA231 and MDA-MB-231? It would also greatly strengthen the paper if the authors can quantify the KD of AXL54xCD3 using Octet and compare it to AXL54.
4. Figure 10B: Very high Caspase 3/7 RLU was unexpectedly observed for HER2-CD3 scFv at ~0.1 pM. Some discussion on this phenomenon would be very helpful.
5. Figure 11A lacks a negative control. It is impressive that both HER2-CD3 and Pro54-CD3 can activate CD8 T cells in the presence of the target cells (MDA231). However, an important criterion for BiTE is that it should not activate T cell in the absence of target cells. A negative control of PBMC incubated with HER2-CD3 or Pro54-CD3 in the absence of MDA231 should be included to demonstrate this.
